# The Multifaceted Roles of MicroRNA-181 in Stem Cell Differentiation and Cancer Stem Cell Plasticity

**DOI:** 10.3390/cells14020132

**Published:** 2025-01-17

**Authors:** Chun Yang, Rui Wang, Pierre Hardy

**Affiliations:** 1CHU Sainte-Justine Research Center, Université de Montréal, Montreal, QC H3T 1C5, Canada; cyang_09@yahoo.com; 2Departments of Pharmacology and Physiology, Université de Montréal, Montreal, QC H3T 1C5, Canada; rui.wang.1@umontreal.ca; 3Departments of Pediatrics, Faculty of Medicine, Université de Montréal, Montreal, QC H3T 1C5, Canada

**Keywords:** human stem cells, cancer stem cells, microRNA-181 (miR-181), epithelial–mesenchymal transition (EMT), cell differentiation, cancer progression, nanotechnology, lipid nanoparticles, therapeutic strategies

## Abstract

Stem cells are undifferentiated or partially differentiated cells with an extraordinary ability to self-renew and differentiate into various cell types during growth and development. The epithelial–mesenchymal transition (EMT), a critical developmental process, enhances stem cell-like properties in cells, and is associated with both normal stem cell function and the formation of cancer stem cells. Cell stemness and the EMT often coexist and are interconnected in various contexts. Cancer stem cells are a critical tumor cell population that drives tumorigenesis, cancer progression, drug resistance, and metastasis. Stem cell differentiation and the generation of cancer stem cells are regulated by numerous molecules, including microRNAs (miRNAs). These miRNAs, particularly through the modulation of EMT-associated factors, play major roles in controlling the stemness of cancer stem cells. This review presents an up-to-date summary of the regulatory roles of miR-181 in human stem cell differentiation and cancer cell stemness. We outline studies from the current literature and summarize the miR-181-controlled signaling pathways responsible for driving human stem cell differentiation or the emergence of cancer stem cells. Given its critical role in regulating cell stemness, miR-181 is a promising target for influencing human cell fate. Modulation of miR-181 expression has been found to be altered in cancer stem cells’ biological behaviors and to significantly improve cancer treatment outcomes. Additionally, we discuss challenges in miRNA-based therapies and targeted delivery with nanotechnology-based systems.

## 1. Introduction

MicroRNAs (miRNAs) are small, non-protein-coding RNA molecules that act as epigenetic regulators suppressing gene expression at the post-transcriptional level [1]. These RNAs range from 19 to 24 nucleotides in length and are highly phylogenetically conserved [2]. MiRNAs interact with complementary sequences in the 3′-untranslated regions of target mRNAs via their seed sequences, and subsequently lead to mRNA degradation or translational suppression. MiRNAs can target multiple mRNAs, thereby modulating several cellular pathways and networks, and playing essential roles in various biological events such as apoptosis, differentiation, angiogenesis, and migration [3]. Substantial evidence indicates key roles for miRNAs in many physiological functions and pathological processes [4,5]. Moreover, miRNA deregulation in various stages of human cancers has been observed to contribute to metastasis and drug resistance. MiRNAs can act as either tumor suppressors or oncogenes, depending on the specific targets that they regulate and the tissues in which they are expressed [6,7].

MiRNA-181 (miR-181) is a multifunctional miRNA involved in numerous biological processes, including proliferation, differentiation, inflammation, and angiogenesis [4,8]. The miR-181 family comprises four highly conserved members (miR-181a/b/c/d) [9], which originate independently from six precursors distributed across three chromosomes. MiR-181a and miR-181b are transcribed from two distinct gene loci: miR-181a-1/miR-181b-1 and miR-181a-2/miR-181b-2 [10,11]. The mature forms of miR-181 family members share the same seed sequence and the sequence similarities and variations among them have been characterized and illustrated by Indrieri et al. [12]. Each miR-181 family member regulates distinct target genes and consequently has diverse functions and context-dependent activities [13]. Emerging studies indicate roles for miR-181 in cellular differentiation in various non-neoplastic diseases, such as osteoarthritis, vascular inflammation, thrombosis, cardiac abnormalities, and pulmonary arterial hypertension. These findings suggest that targeting specific members of the miR-181 family might provide a promising therapeutic approach for treatment or tissue repair in various human diseases [8,14,15]. Additionally, miR-181 members are dysregulated in various tumor tissues and display either tumor-suppressive or oncogenic properties, depending on the specific cancer type [4,16,17]. MiR-181 plays key roles in the development and metastasis of several human cancers, including breast cancer (BC), hepatocellular carcinoma (HCC), and human gastric cancer [18,19].

Stem cells are human cells capable of self-renewal and differentiation into various cell types [20]. Embryonic stem cells (ESCs) and adult stem cells (ASCs, also termed somatic stem cells) are two main types of human stem cells. ESCs are derived from the blastocyst inner cell mass and can differentiate into all cell types of the three germ layers. ASCs are tissue-specific undifferentiated cells named according to their tissue of origin. Examples include epithelial stem cells, hematopoietic stem cells (HSCs, or blood stem cells), and mesenchymal stem cells (MSCs). Currently, ASCs are used in clinical tissue and organ repair because of their wide availability, low tumorigenicity, and low risk of transplant rejection. Induced pluripotent stem cells (iPSCs) are a new type of pluripotent cells that can be generated from adult somatic cells, such as skin fibroblasts or peripheral blood mononuclear cells, through genetic reprograming or the “forced” introduction of embryonic genes (Oct4, Sox2, Klf4, and c-Myc) [21]. The proteins encoded by these genes aid in reprogramming somatic cells into a state similar to that of ESCs. The transcription factor OCT4 is expressed exclusively in undifferentiated ESCs and in all pluripotent cells during embryogenesis, in which it plays crucial roles in establishing and maintaining cellular pluripotency [22,23]. The SRY-related high-mobility group-box (SOX) transcription factor family plays roles in regulating embryonic development, maintaining stemness, and controlling cell differentiation. SOX2 collaborates with OCT4 in regulating the expression of genes such as *Fgf4* and *Nanog* [24]. The transcription factor KLF4 regulates proliferation, differentiation, apoptosis, and somatic cell reprogramming [21]. NANOG is a DNA-binding homeobox protein that helps ESCs maintain pluripotency by suppressing cell determination [23,25]. C-Myc regulates various biological processes, including the balance between stem cell differentiation and self-renewal [26,27]. Overall, stem cells have substantial potential for clinical therapy because of their ability to self-renew and pluripotency [28].

## 2. MiR-181 in Stem Cell Differentiation

ESCs retain a self-renewal capability, thereby maintaining stem cell characteristics and pluripotency to produce different cell progeny through differentiation [29]. The key signaling pathways supporting the self-renewal and pluripotency of human ESCs involve transforming growth factor-beta (TGF-β)/Smad, the insulin-like growth factor and fibroblast growth factor receptor-regulated AKT and MAPK networks, and wingless type MMTV integration site (Wnt)/β-catenin; these pathways promote the expression of ESC-specific (pluripotency) genes such as Sox2, Oct-4, and Nanog. In contrast, the Notch and bone morphogenetic protein signaling pathways upregulate differentiation-specific genes and consequently drive the differentiation of ESCs [30,31]. MiRNAs play fundamental roles in regulating ESC fate decisions by controlling the balance between self-renewal and differentiation [32,33]. The MiR-181 family is widely recognized for its broad pathophysiological effects on cell fate and function, including facilitating ESC differentiation [4]. The miR-181 family members are highly expressed in differentiated human ESCs, in which they regulates coactivator-associated protein arginine methyltransferase 1 (CARM1) [34], a protein with a key role in regulating pluripotency in human ESCs [35]. Xu et al. have found that miR-181c is highly expressed and strongly represses CARM1 expression induced during ESC differentiation. Enforced expression of miR-181c in undifferentiated ESCs inhibits the expression of pluripotency genes, thereby inducing differentiation [34]. Furthermore, Polycomb repressive complexes (PRCs) act as key regulators of the pluripotency and differentiation of ESCs and ASCs by modifying the chromatin architecture and maintaining gene repression [36]. Mammalian genomes encode multiple homologs of PRC1 components. O’Loghlen et al. have identified chromobox homolog 7 (CBX7) as the primary Polycomb ortholog of PRC1 complexes in ESCs and have found that CBX7 represses its homologs. During ESC differentiation, the overexpression of miR-181a or miR-181b regulates the PRC1 composition by suppressing CBX7 expression, thereby alleviating its repression of CBX7 homologs [37].

MSCs are typically defined as plastic-adherent, fibroblast-like cells found in various tissues, including bone marrow (BM), adipose tissue, and the umbilical cord. MSCs have the multipotent potential to differentiate into specialized cells, such as adipocytes, chondrocytes, osteoblasts, myocytes, and neurons [38]. Similar to ESCs, in MSCs, differentiation is controlled by the TGF-β, bone morphogenetic protein, Notch, hedgehog, Wnt, fibroblast growth factor, and epidermal growth factor signaling pathways [39]. MSCs and their derived extracellular vesicles (EVs) have therapeutic effects in regenerative medicine [40,41], tumor growth/inhibition [42], and immunoregulation [43]. The ability of MSCs to differentiate into multiple cell types makes them a promising source for tissue repair approaches. BM contains a population of multipotent BM-MSCs, which are most frequently used in regenerative medicine because of their multipotent properties and high proliferative capability [44,45]. Although the ability of human MSCs to differentiate into various mesenchymal cell lineages renders them particularly suitable for tissue repair strategies, the differentiation potential of human MSCs varies among donors [46]. To identify biomarkers indicating donors with high differentiation potential, Georgi et al. have investigated the miRNA expression levels of high-potential and low-potential BM-MSCs undergoing chondrogenesis. An analysis of miRNA expression during a 7-day differentiation period indicated that miR-181 showed elevated expression in high-potential MSCs and was also upregulated in low-performing MSCs during chondrogenesis, thus suggesting that miR-181 is involved in the negative regulation of chondrogenesis [47]. Wang et al. have reported a critical role for miR-181 in inducing the osteogenic differentiation of BM-MSCs. Overexpression of miR-181 decreases the expression of lysine acetyltransferase 2B (KAT2B), thus enhancing the osteogenic differentiation capability of BM-MSCs [48]. KAT2B, a member of the lysine acetyltransferase family, has been implicated in regulating acetylation and transcription levels in diverse biological processes [49]. Additionally, BM-MSCs are a key component of the hematopoietic microenvironment and support hematopoiesis [38,50]. In mouse BM, miR-181 is preferentially expressed in B cells compared with undifferentiated progenitor cells or other lineages. Overexpression of miR-181 in HSCs has been found to increase the fraction of B-lineage cells in both in vivo and in vitro studies [14], thus suggesting that miR-181 promotes hematopoietic lineage differentiation. Lin28 is a pluripotency marker that contributes to maintaining cell pluripotency, partly by inhibiting the maturation of let-7 miRNAs [51,52,53,54]. MiR-181 overexpression and inhibition studies have proven its role in promoting megakaryocyte (MK) differentiation by suppressing the expression of Lin28 in human CD34+ HSCs [54]. In an investigation of miRNAs’ roles in the lineage differentiation of amniotic fluid stem cells (AFSCs), Iordache et al. have observed that miR-181 is upregulated in endothelial progenitor cells derived from AFSCs [55]; therefore, this miRNA might have a role in lineage-specific differentiation (Table 1).

In contrast, Su et al. have demonstrated an inhibitory effect of miR-181a on the differentiation of CD34+ HSCs into granulocytes and macrophages by downregulating targets such as calcium/calmodulin-dependent protein kinase, small phosphatase like, and protein kinase C delta CTD. This downregulation affects the protein kinase C delta-P38-C/EBPα pathway by decreasing the phosphorylation of retinoblastoma protein (a tumor suppressor involved in cell cycle regulation). In a xenograft mouse model of acute myeloid leukemia, miR-181a inhibition has been found to enhance myeloid differentiation from CD34+ HSCs and alleviate leukemic symptoms [56]. In agreement with these findings, De Luca et al. have shown that the overexpression of miR-181a-5p in umbilical cord blood hematopoietic CD34+ stem cells (a source of HSCs used for various treatments) increases cell viability while decreasing differentiation [57]. Judson et al. have observed a transient elevation in the expression of miR-181 family members during the dedifferentiation of mouse fibroblasts to iPSCs. The expression of miR-181 is regulated by the core reprogramming cocktail OSK (consisting of the transcription factors Oct4, Sox2, and Klf4) [58], whereas inhibiting endogenous miR-181 decreases iPSC colony formation. Thus, miR-181 has been identified as a novel enhancer of reprogramming, partly through the suppression of nuclear receptor subfamily 2 group C member 2 (*Nr2c2*) and myristoylated alanine-rich c-kinase substrate (*Marcks*) [59]. Both NR2C2 and Marcks play fundamental roles in early embryonic development and stem cell function [60,61].

**Table 1 cells-14-00132-t001:** Roles of miR-181 family members in normal stem cell differentiation.

Cells/Tissues	Upstream	MiR-181	Targets	Differentiation/Stemness	Ref.
HSCs		miR-181		Promotes HSC differentiation to B-lineage cells	[14]
ESCs		miR-181s, miR-181c	CARM1	Induces the differentiation of human ESCs	[34]
ESC differentiation		miR-181, miR-181a/b	CBX7	Accelerates ESC differentiation	[37]
BM-MSCs	DOT1L	miR-181	KAT2B/SRSF1	Induces the osteogenic differentiation of BM-MSCs	[48]
MK hematopoiesis		miR-181	Lin28-let-7	Promotes MK hematopoiesis	[54]
Endothelial progenitor cells derived from AFSCs		miR-181a-5p		Promotes the differentiation of AFSCs	[55]
BM-MSC-derived EVs		miR-181a-5p	EGR2	Inhibits the differentiation of cord blood HSCs	[57]
iPSCs	OSK	miR-181	Nr2c2, Marcks	Enhances iPSC reprogramming	[59]

## 3. Cancer Stem Cells and the Oncogenic Epithelial–Mesenchymal Transition

ESCs have an extraordinary capability to produce diverse cell progeny through a sequential epithelial–mesenchymal transition (EMT) process. The EMT is a cell biological program that occurs during development, adult tissue regeneration, wound healing, and fibrosis in adult tissues [62,63]. However, under pathological conditions, the EMT can confer stemness or stem-like phenotypes, thereby serving as a major mechanism for generating cancer stem cells (CSCs) [64]. CSCs, also known as tumor-initiating cells, are a subpopulation of cancer cells derived from either differentiated cancer cells or somatic stem cells [64,65,66]. Despite being a minor tumor component, CSCs play fundamental roles in mediating the recurrence of tumors after therapy, tumor dormancy, metastasis, and chemoresistance [67]. Extensive evidence indicates that almost all treatment-naive tumors contain CSCs. Notably, ESCs and CSCs share several common features, such as pluripotency, self-renewal, the expression of stemness-associated genes, and the acquisition of EMT traits [68].

The EMT is a reversible transdifferentiation process in which epithelial cells gain mesenchymal characteristics. The substantial phenotypic changes during the EMT include loss of polarity and cell–cell adhesion, along with the acquisition of migratory and invasive properties. The EMT program consists of the sequential activation of several intracellular signaling pathways, notably those mediated by TGF-β, Wnt/β-catenin, Notch, Hedgehog, and receptor tyrosine kinases [69,70]. The EMT is also regulated by EMT-inducing transcription factors (EMT-TFs), such as zinc finger E-box-binding homeobox (ZEB1), ZEB2 (also known as SIP1), Snail, Slug, E12/E47, Kruppel-like factor 8 (KLF8), and Twist [71]. Additionally, the SOX family members SOX2, SOX9, and SOX17 act synergistically with Snail family transcriptional repressors in influencing the EMT process [72]. During the EMT, epithelial markers such as E-cadherin, claudin, ZO-1, laminin-1, type IV a1-collagen, and cytokeratin are progressively downregulated. In contrast, mesenchymal markers, including N-cadherin, vimentin, smooth muscle actin, and fibronectin, are upregulated. E-cadherin is a critical epithelial marker essential for the formation of adherens junctions. A loss of E-cadherin causes major changes in cell physiology, including enhanced migratory and invasive behaviors [69,71]. Vimentin, a mesenchymal marker, plays crucial roles in regulating the expression of many EMT-associated genes, stabilizing pro-EMT pathways, and promoting cell migration and metastasis [73]. As the EMT progresses, cell–cell junctions are disrupted, the actin cytoskeleton undergoes extensive reorganization, and cells acquire heightened motility and invasive capabilities (Figure 1).

The EMT process is integral to cancer progression and has been identified to varying degrees across multiple human cancer types, including pancreatic cancer, prostate cancer, colorectal cancer, ovarian cancer, and oral squamous cell carcinoma (SCC) [74,75,76,77,78,79,80,81,82,83]. A strong connection has been established between the EMT in cancers and the acquisition of a stem-cell-like state. EMT activation enhances tumor cells’ migratory and invasive properties, which are critical for tumor dissemination. In contrast, the reverse process, the mesenchymal-to-epithelial transition (MET), is essential for the development of metastatic tumors at distant sites. A key hallmark of the EMT is the resistance of tumor cells to anoikis, a form of programmed cell death triggered by detachment from the extracellular matrix. This resistance is also a defining feature of CSCs [75,84]. CSCs simultaneously express epithelial and mesenchymal phenotypic traits, and consequently can transition between states during the EMT process [64,85]. In human mammary epithelial cell cultures, a subpopulation of CSCs undergoes the EMT and forms mammospheres (MSs) upon stimulation with TGF-β or ectopic expression of EMT-TFs such as Snail, or Twist [86]. Despite the dynamic and transient nature of the EMT, CSCs often exhibit mesenchymal-like characteristics. The development of CSCs and the EMT process share common signaling pathways, including the Wnt, Notch, and hedgehog pathways [87,88]. Many EMT-TFs, such as N-cadherin, Snail1/2, and ZEB1/2, play essential roles in driving CSC features. These factors underscore the close relationships among the EMT, stemness, and cancer progression.

## 4. MiR-181 in the EMT and CSCs

The process of CSC formation is regulated by a combination of genetic and epigenetic factors. Emerging evidence indicates that abnormal miRNA expression contributes to the initial formation of CSCs and leads to dysregulated self-renewal and cancer progression [89]. Several studies have demonstrated that miRNAs play major roles in metastatic tumor progression by modulating the reversible EMT process through the control of EMT-related signaling pathways and targeting of stemness-related factors [90]. Several miRNAs have been shown to influence CSC initiation, maintain stemness properties, regulate CSC function, and act as E-cadherin repressors by targeting multiple oncogenic pathways, such as Notch, PI3K/AKT, WNT/β-catenin, JAK/STAT, and nuclear factor kappa B (NF-κB) [90,91,92,93]. Among these miRNAs, miR-181 has garnered particular attention for its involvement in the EMT process. Polo-Generelo et al. have identified a TGF-β-responsive long non-coding RNA (lncRNA), *Lnc-Nr6a1*, expressed during the EMT in mouse mammary epithelial cells. The sequences of pri-miR181a2 and pri-miR181b2 are found in a non-polyadenylated isoform of *lnc-Nr6a1* and give rise to mature miR-181a2 and miR-181b2 after processing by Dicer. These miRNAs enhance the cell invasive capability and confer resistance to anoikis [94]. Interestingly, miR-181b exhibits tissue-specific roles in EMT regulation. In mouse airway tissues, miR-181b suppresses the EMT, a key event in airway remodeling. Huang et al. have demonstrated that the overexpression of miR-181b decreases the expression of EMT-associated factors such as vimentin and α-SMA while increasing E-cadherin expression. This regulation occurs through the direct targeting of the high mobility group box 1 (HMGB1) mRNA, thus enhancing the activation of NF-κB signaling. Furthermore, miR-181b is sponged by the lncRNA TUG1, which contributes to airway remodeling [95]. The identification of miR-181 family dysregulation in various tumor tissues highlights the critical roles of these family members in both CSCs and the EMT. These findings underscore the diverse and context-dependent functions of miR-181, which are closely associated with cancer progression [96]. A summary of the roles of this miRNA in CSCs and the EMT is provided below (Table 2).

### 4.1. MiR-181 in Breast Cancer

BC is the most prevalent cancer globally. Metastatic BC is the most advanced BC stage and is associated with high mortality rates [97]. The EMT process is critical in the initiation and progression of BC metastasis by facilitating the transformation of non-invasive BC to invasive BC [98]. TGF-β-dependent signaling has been strongly implicated in promoting the EMT during advanced stages of BC [99,100]. Wang et al. have demonstrated that TGF-β induces the expression of miR-181 and increases the population of BC cells capable of forming MSs in suspension culture, a hallmark of stem cell-like properties. Notably, miR-181 family members are highly expressed in MSs cultured under undifferentiated conditions [101]. Ataxia telangiectasia mutated (ATM), a serine/threonine kinase, is essential for DNA damage-induced cellular responses [102]. The downregulation of ATM by the overexpression of miR-181a/b induces a sphere-forming CSC phenotype in BC cells [101]. Using the overexpression of miR-181a and the miR-181a sponge (miArrest 181a), Taylor et al. have identified miR-181a as a TGF-β-regulated “metastamir” that drives the metastatic potential of BCs by promoting the EMT, migration, and invasion. The expression of miR-181a is selectively upregulated in metastatic BC, particularly in triple-negative BC tumors. High levels of miR-181a are associated with pulmonary micrometastatic outgrowth and lethality in mice with late-stage BC. Mechanistically, miR-181a suppresses the expression of the pro-apoptotic protein Bim, thereby decreasing the sensitivity of metastatic cells to anoikis [103]. Moreover, the expression of miR-181b is upregulated by high mobility group A1 (HMGA1), a protein with a critical role in the TGF-β signaling network that drives the EMT and maintains the undifferentiated phenotype of CSCs [104]. HMGA1 upregulates miR-181b, which in turn suppresses the tumor suppressor CBX. MiR-181b overexpression contributes to the EMT and BC progression [104,105]. Yoo et al. have further demonstrated through miR-181b-3p overexpression and inhibition experiments that miR-181b-3p promotes the Snail-induced EMT and subsequent activation of MCF-7 BC cell metastasis [106]. By directly targeting the oncogenic signaling adaptor protein YWHAG, miR-181b-3p stabilizes Snail and consequently leads to EMT-associated morphological changes, increased invasiveness, and altered expression of EMT markers in metastatic BC cells [106,107].

In contrast, Kastrati et al. have revealed that miR-181 overexpression inhibits estrogen receptor (ER)+ BC tumor growth by downregulating pleckstrin homology-like domain, family A, member 1 (PHLDA1), a key protein in stem cell maintenance and cell survival [108]. PHLDA1 is highly expressed in MSs of ER+ BC cells, and its inhibition impairs MS formation and decreases the population of aldehyde dehydrogenase (ALDH)-positive cells. ALDH activity is a frequently used marker for stem cells. Interestingly, miR-181 expression is suppressed by E2 and tumor necrosis factor α (TNFα) in an ER- and NF-κB-dependent manner. Moreover, TNFα promotes BC invasion through EMT programs, thus further repressing E-cadherin and enhancing tumor aggressiveness [109].

### 4.2. MiR-181 in Glioblastoma Multiforme

Glioblastoma multiforme (GBM), a highly malignant astrocytoma, is among the most aggressive forms of glioma. In GBM, a small population of GBM stem cells (GSCs) acts as a reservoir for tumor recurrence and progression. The mesenchymal reprogramming of GSCs within the tumor microenvironment is a critical determinant of GBM outcomes [110]. The stemness-associated transcription factor Notch2 is highly expressed in GBM, and its activation plays critical roles in GBM formation and progression [111]. Although miR-181a downregulates Notch2 expression, miR-181a itself is downregulated in GSCs derived from U87 and U373 human glioblastoma cells. Overexpression of miR-181a inhibits the expression of stemness-associated markers such as CD133 and BMI1, thereby decreasing the tumorigenicity of GSCs [112]. The miR-181 family inversely correlates with NF-κB-targeting gene expression and the activity of EMT pathways in GBM. This miRNA family has been shown to inhibit glioblastoma cell invasion and proliferation [113]. Wang et al. have demonstrated that the upregulation of the miR-181 family reverses the EMT by targeting karyopherin subunit alpha 4 (KPNA4), a gene whose protein product activates NF-κB-associated pathways critical for the EMT in multiple cancer types [113,114]. Among miR-181 family members, miR-181b exhibits the most potent inhibitory effects on the EMT in glioblastoma [113]. He et al. have reported the EMT-inhibitory role of miR-181c in the glioblastoma EMT [115]. MiR-181c is often downregulated in glioblastoma, whereas its overexpression leads to EMT suppression. Forced overexpression of miR-181c increases E-cadherin levels and decreases the levels of mesenchymal markers, including N-cadherin and vimentin. Additionally, miR-181c inhibits TGF-β signaling in GBM cells, thus further curbing the EMT and tumor aggressiveness [115].

### 4.3. MiR-181 in Hepatocellular Carcinoma

HCC constitutes more than 90% of liver cancers. CSCs in HCC are identified through the expression of several surface antigens, including CD133, CD90, CD44, OV6, and epithelial cell adhesion molecule (EpCAM), as well as Hoechst staining of side population cells [116]. Stem cell activators such as the Notch, TGF-β, Wnt/β-catenin, EpCAM, Lin28, and Hedgehog signaling pathways drive HCC progression by activating CSCs [117]. Among these, TGF-β, a key driver of the EMT, plays a crucial role in HCC pathogenesis [118]. The miR-181 family has been widely implicated in mediating TGF-β signaling and the EMT in HCC [119,120]. Brockhausen et al. have reported the significant upregulation of miR-181a during the TGF-β-induced EMT in hepatocytes. Overexpressed miR-181a in the mouse liver leads to genetic alterations associated with TGF-β signaling and the EMT, thus highlighting its roles in the hepatocyte EMT and TGF-β-mediated effects [119]. Similarly, Riccioni et al. have corroborated the involvement of miR-181 family members in the EMT in hepatocytes and HCC cells [121]. Heterogeneous nuclear ribonucleoprotein Q (hnRNP-Q) has been identified as a “mesenchymal” gene in hepatocytes. During EMT/MET dynamics, miR-181-a1-3p and miR-181-b1-3p are among the miRNAs affected by HnRNP-Q knockdown [121]. In liver cancers induced by diethylnitrosamine, miR-181 expression increases alongside tumor growth [122]. However, the liver-specific deletion of miR-181ab1 (mir181a-1-mir181b-1 cluster) upregulates the tumor suppressor CBX7, a confirmed miR-181 target, and inhibits liver tumor initiation and progression. This deletion also increases E-cadherin expression and partially reverses the EMT [122].

Additionally, several independent studies have defined the role of miR-181 in malignant hepatic stemness. (1) Arzumanyan et al. have demonstrated that hepatitis B virus-encoded X antigen (HBx) upregulates miR-181, along with Oct-4, Nanog, Klf-4, β-catenin, and EpCAM, thus promoting self-renewal in HCC CSCs [123]. (2) High miR-181 expression has been observed in embryonic livers, isolated hepatic stem cells, and HCC metastases. MiR-181 expression is regulated by the Wnt signaling pathway [11]. In some cases, miR-181 stimulates Wnt/β-catenin signaling by downregulating GSK3b signaling [10]. Ji et al. have shown that forced expression of miR-181 enhances the tumor-initiating ability of EpCAM+ HCC cells by targeting hepatic differentiation regulators, such as caudal type homeobox 2 (CDX2), GATA binding protein 6 (GATA6), and nemo-like kinase (NLK, a Wnt/beta-catenin pathway inhibitor) [10]. (3) MiR-181 expression is influenced by IL-6 and Twist in hepatocellular CSCs. Twist, a known inhibitor of myogenic differentiation, promotes motility and invasion in these cells. Meng et al. have demonstrated that the overexpression of miR-181a/b inhibited Ras association domain family 1A (RASSF1A, a tumor suppressor), tissue inhibitor of metalloprotease 3 (TIMP3), and NLK protein expression in hepatocellular CSCs, indicating that miR-181 mediates the oncogenic effects of Twist on the invasion of these cells [124]. In summary, miR-181 plays critical roles in the regulation of the EMT, CSC activation, and tumor progression in HCC, and mediates key oncogenic pathways.

### 4.4. MiR-181 in Other Types of Human Cancer

Independent studies have highlighted the anti-CSC properties of miR-181 in various cancers, including SCC, melanoma, and non-small cell lung cancer (NSCLC).

To date, two studies have suggested anti-CSC effects of miR-18 on SCC cells. In human papillomavirus 16 (HPV16)-transfected oral and oropharyngeal SCC cells, miR-181a and miR-181d (miR-181a/d) expression are suppressed by HPV16 [125]. The ectopic expression of miR-181a/d in HPV16-transfected oral and oropharyngeal SCC cells decreases anchorage-independent growth and the CSC-like phenotype by targeting K-ras and ALDH1 [125]. In cervical SCC, miR-181c-5p directly downregulates the expression of glycogen synthase kinase 3 beta interacting protein (GSKIP) [126], an A-kinase anchoring protein involved in Wnt signaling regulation [127]. Overexpression of miR-181c-5p inhibits the stem-like properties of cervical SCC (SiHa) cells by decreasing the expression of stem cell markers (SOX2, OCT4, CD44, N-cadherin, and vimentin) and increasing E-cadherin levels [126].

In melanoma stem cells, miR-181a induces apoptosis by directly targeting Bcl-2. Zhang et al. have identified that the lncRNA LHFPL3-AS1-long interacts with miR-181a-5p, thus preventing the miR-181a-5p-mediated degradation of the Bcl-2 mRNA and suppressing melanoma stem cell death [128].

In NSCLC cells, ectopic expression of miR-181b decreases the CD133+ population, inhibits CSC-like properties, and enhances sensitivity to cisplatin treatment by targeting Notch2, thereby inactivating the Notch2/Hes1 signaling pathway [129]. Increased expression of miR-181b suppresses stem cell-like characteristics, such as sphere formation, and markedly downregulates stemness-associated transcription factors (Notch2, NICD2, HES1, and HEY1). However, miR-181b inhibition increases Notch2 expression and has a significant relationship with the overall survival and CSC-like properties of NSCLC patients. [129].

An unappreciated role of miR-181a has been identified in high-grade serous ovarian cancer, a leading cause of cancer-associated deaths among women. Parikh et al. have reported that the ectopic expression of miR-181a promotes the TGF-β-mediated EMT by directly targeting Smad7 [130]. Smad7, an inhibitor of TGF-β signaling in epithelial ovarian carcinoma, helps maintain the epithelial phenotype through the MET [131]. Conversely, miR-181a inhibition via decoy vector suppression results in significant reversion of these phenotypes [130]. In a study of HT-29 human colon adenocarcinoma cells, Cai et al. developed an EMT model by treating the cells with TGF-β. In this model, a collection of miRNAs was found to be dynamically regulated, including miR-181, which was significantly downregulated [132]. However, in colorectal cancer, one of the most prevalent cancers worldwide [133], miR-181a is significantly upregulated in cancerous tissues with liver metastasis. Overexpression of miR-181a not only promotes colorectal cancer cell motility, invasion, and liver metastasis, but also plays a potential role in enhancing the EMT by upregulating vimentin and downregulating epithelial markers such as β-catenin and E-cadherin through the suppression of the Wnt inhibitory factor 1 (WIF-1) [134]. WIF1, a tumor suppressor gene, regulates tumor invasion through the EMT process [135].

**Table 2 cells-14-00132-t002:** Regulatory roles of the miR-181 family in the EMT and CSCs of human cancers.

Cells/Tissues	Upstream	MiR-181	Targets	Stemness/EMT	Refs.
Metastatic BC, MDA-MB-361	TGF-β	miR-181a, miR-181b	ATM	Induces the sphere-forming CSC phenotype in BC cells	[101]
Metastatic BC, triple-negative BC	TGF-β	miR-181a	Bim	Promotes the EMT	[103]
BC, MCF-7	HMGA1	miR-181b	CBX7	Induces the EMT of endometrial epithelial cells	[104]
BC, MCF-7		miR-181b-3p	YWHAG/Snail	Promotes EMT-characteristic morphological changes	[106]
ER^+^ BC, MCF-7	E2, TNFα	miR-181a	PHLDA1	Inhibits stem-like properties, impairs MS formation	[108]
GBM		miR-181a	Notch2	Inhibits stemness-associated markers (CD133 and BMI1) and the tumorigenicity of GSCs	[112]
GBM		miR-181s, miR-181b	KPNA4	Reverses the EMT	[113]
GBM		miR-181c	N-cadherin, vimentin, TGF-β	Inhibits the EMT of GBM	[115]
Hepatocyte EMT	TGF-β	miR-181a		Induces the hepatocyte EMT	[119]
Hepatocytes and HCC cells	SYNCRIP	miR-181a1-3p, miR-181b1-3p		Promotes the EMT	[121]
miR-181ab1 deficient liver tumors		miR-181ab1	CBX7	Promotes the EMT	[122]
Liver CSCs	HBx	miR-181		Maintains “stemness”	[123]
Liver CSCs, EpCAM+ hepatic CSCs		miR-181	CDX2, GATA6, NLK	Promotes the stem-cell-like features of HCC cells	[10]
Hepatocellular CSCs	IL-6, Twist	miR-181a/b	RASSF1A, TIMP3, NLK	Induces hepatocellular CSC motility and invasion	[124]
Oral/oropharyngeal SCC	HPV16	miR-181a/d	K-ras, ALDH1	Inhibits the CSC phenotypes of HPV-16-transfected oral/oropharyngeal SCC cells	[125]
Cervical SCC		miR-181c-5p		Inhibits the stem-like properties of cervical SCC cells	[126]
Melanoma stem cells	lncRNA LHFPL3-AS1-long	miR-181a-5p	Bcl-2	Induces melanoma stem cell death	[128]
NSCLC		miR-181b	Notch2, NICD2, HES1, HEY1	Attenuates CSC characteristics, decreases the CD133+ population	[129]
Ovarian cancer		miR-181a	Smad7	Promotes the TGF-β-mediated EMT	[130]
Colorectal cancer		miR-181a	WIF-1	Promotes the EMT	[134]

The roles of miR-181 in CSCs and the EMT in cancer cells are complex. Generally, studies have shown inhibitory effects of this miRNA on glioma, SCC, and NSCLC, but have highlighted its promotion of the EMT and CSCs in BC, hepatocarcinoma, ovarian cancer, and colorectal cancer. The contradictory effects of miR-181 on the EMT and CSCs across studies might stem from variations in the tissue origin, clinical stage distribution, metastatic status, and analytical methods. Although TGF-β is widely recognized as an inducer of the EMT and miR-181 targets several key molecules within the TGF-β signaling pathway in various human cells, the precise molecular mechanisms governing the interaction between TGF-β and miR-181 and their involvement in the EMT remain incompletely understood. Advancing our understanding of how miR-181 regulates the EMT and CSCs might pave the way for new approaches in cancer diagnosis and prognosis. Furthermore, identifying potential therapeutic targets of miR-181 could potentially provide strategies to inhibit metastatic dissemination and improve patient survival.

## 5. Therapeutic Potential of miR-181

Recent studies have highlighted the roles of miRNAs in various biological processes, including cell differentiation and tissue regeneration. Therefore, miRNAs are promising candidates for optimizing tissue repair and regeneration strategies. For instance, Qi et al. encapsulated 181a/b-1 into poly (lactic-co-glycolic acid) (PLGA) nanofibers, a highly effective biodegradable polymer. Their findings revealed that PLGA nanofibers loaded with miR-181a/b-1, compared with non-encapsulated miR-181a/b-1, significantly increase the osteogenic differentiation of human adipose-derived MSCs [136]. These advancements suggest a promising direction for the development of miRNA-based regenerative therapies.

The EMT is a critical process driving invasion and metastasis during cancer progression, as reviewed by Kang et al. [71]. CSCs play critical roles in tumor relapse and metastatic tumor growth, and are known contributors to cancer resistance and poor clinical outcomes. Because both the EMT and CSC are closely linked and substantially contribute to cancer recurrence and metastasis, targeting these processes represents a compelling therapeutic strategy. MiRNAs regulate CSC stemness by directly targeting stemness-associated TFs and markers or by indirectly reversing the EMT [137]. To restore the downregulated miRNAs in CSCs or cancer cells, miRNA-specific mimics can re-establish miRNA levels and restore their associated biological functions. In contrast, approaches such as anti-miRNA oligonucleotides, genetic knockout technologies, and miRNA sponges effectively suppress the aberrantly elevated expression of miRNAs [138,139,140,141]. These interventions make stemness-associated miRNAs valuable as diagnostic and prognostic biomarkers, as well as targets for cancer therapy. Notably, substantial evidence links the EMT process to miR-181 regulation [96]. Whereas the functions of miR-181 family members and their target genes vary by cancer type and gene expression profiles, targeting miR-181-associated signaling pathways in CSCs and in the EMT process has substantial therapeutic potential. In certain cancers, miR-181 inhibition might deplete CSCs by inducing their differentiation. However, further research is essential to deepen our understanding of miR-181’s role in CSCs hierarchies and to design more effective and specific anti-CSC therapies [116].

## 6. Current Progress of miRNA-Based Therapies

Despite their therapeutic potential, miRNAs face major challenges in clinical application, primarily related to their stability and sustainability in the circulation, safety and toxicity, and delivery. Unmodified, naked miRNAs are rapidly degraded by RNase A in the bloodstream and cleared through renal excretion. Additionally, miRNA delivery faces hurdles such as low endocytosis, potential cytotoxic effects on healthy tissues, and immunotoxicity, which complicate therapeutic development [142]. For example, MRX34 (miR-34a mimic, Mirna Therapeutics) was designed to restore the tumor-suppressive function of miR-34a, which was tested in a Phase I clinical trial for patients with advanced solid tumors [143]. However, the trial was halted in 2016 due to dose-limiting toxicity and severe immune-related side effects. Additionally, Miravirsen (Santaris Pharma), a locked nucleic acid inhibitor targeting miR-122, showed promising results in Phase II clinical trials by reducing the viral load in hepatitis C virus-infected patients. However, safety concerns related to long-term inhibition of miR-122 have raised questions about its broader therapeutic use [144]. MiRNA-based therapies are still in the experimental phase, with some notable failures, but they hold significant promise for diseases with complex genetic regulation.

To overcome these challenges, chemical modifications have been introduced to enhance the stability of therapeutic miRNAs [145]. For example, 2′-ribose modified miRNAs have enhanced stability and a prolonged half-life in the systemic circulation [146]. Beyond viral delivery, nanotechnology-based delivery systems have been designed to encapsulate miRNAs [4,145,147]. Nanoparticles (NPs) not only enhance miRNA stability in the serum but also improve endosomal escape and biocompatibility. Surface functionalization of NPs with targeting moieties enables specific binding to biomolecules expressed on CSCs or cancer cells, thus improving therapeutic targeting [148,149]. Among NP platforms, lipid nanoparticles (LNPs) are particularly promising delivery vectors for nucleic acids [141,150]. LNPs effectively encapsulate miRNAs, thereby protecting them from nuclease degradation and overcoming physiological barriers to the delivery of miRNAs to difficult-to-reach tissues [151]. Notable examples include (1) the LNP formulation of siRNAs targeting transthyretin approved by the FDA [152], (2) nucleoside-modified mRNA-LNP vaccines successfully developed during the COVID-19 pandemic [153], and (3) an LNP platform encapsulating an miR-193a-3p mimic for solid tumors, which has reached a Phase 1 trial (NCT04675996) [154]. Our research team has successfully encapsulated an miR-181a mimic using liposomes and solid LNPs, and investigated their anti-neoplastic effects in preclinical studies as treatments for retinoblastoma and glioblastoma [155,156]. These encapsulation methods enable effective miR-181a delivery to tumors and the subsequent suppression of tumor growth. Notably, hyaluronic acid-conjugated LNPs have been used to selectively deliver miR-181a to CD44-positive cells, including CSCs [156].

Notably, EVs represent a promising alternative for delivering genetic therapeutics. EVs are small lipid-enclosed particles including microvesicles, exosomes, ectosomes, membrane vesicles, and apoptotic bodies [157,158]. Compared with artificial NPs, EVs offer advantages such as lower immunogenicity and toxicity, and the ability to cross plasma membranes and diffuse into tumor tissues [159]. Recent studies have highlighted the immunomodulatory effects of MSC-derived exosomes, which can modulate the local microenvironment [159,160,161,162]. MiR-181 is notably abundant in MSC-derived exosomes and exhibits substantial anti-inflammatory regulatory functions [162]. For example, human umbilical cord MSC-derived exosomes with high miR-181c levels have been found to decrease burn-induced inflammation by downregulating Toll-like receptor 4 and its pathway [163,164]; moreover, BM MSC-derived exosomes rich in miR-181c have been found to inhibit inflammation and apoptosis, thus alleviating spinal cord injury [165]. Although the effects of miR-181 in MSC-derived exosomes on CSC differentiation or the EMT remain unclear, its diverse therapeutic roles underscore a need for further investigation of its clinical potential. Continued research might uncover novel therapeutic strategies for addressing a range of diseases.

## 7. Conclusions

The miR-181 family plays critical roles in regulating human stem cells by contributing to their formation, maintenance, and differentiation through the targeting of key stem cell-related signaling pathways in various human stem cell types. These miRNAs are also intricately associated with nearly all EMT-associated signaling pathways, and substantially influence cancer cell behavior through EMT and CSC modulation. The effects of miR-181 vary according to disease-specific cellular contexts and tissue-specific molecular signaling pathways, thus showcasing its therapeutic versatility beyond cancer treatment. Given its diverse functions, miR-181’s immense therapeutic potential warrants further investigation. Advances in genome-wide screening technologies, including single-cell and spatial multi-omics technologies, should provide promising tools to identify new miR-181 targets and pathways involved in CSC regulation. These advancements might clarify the role of miR-181 in stem cell biology and facilitate its application in therapeutic strategies. Although the rapid development of miRNA-based therapy holds great promise, challenges such as off-target effects and potential adverse effects persist. NP delivery platforms have successfully delivered anti-miR-181a oligonucleotides or miR-181 mimics in preclinical models [155,156,166,167]. In the future, combining miR-181-based targeted therapy with advanced nanodelivery systems might pave the way for breakthroughs in stem cell therapy and other regenerative treatments.

## Figures and Tables

**Figure 1 cells-14-00132-f001:**
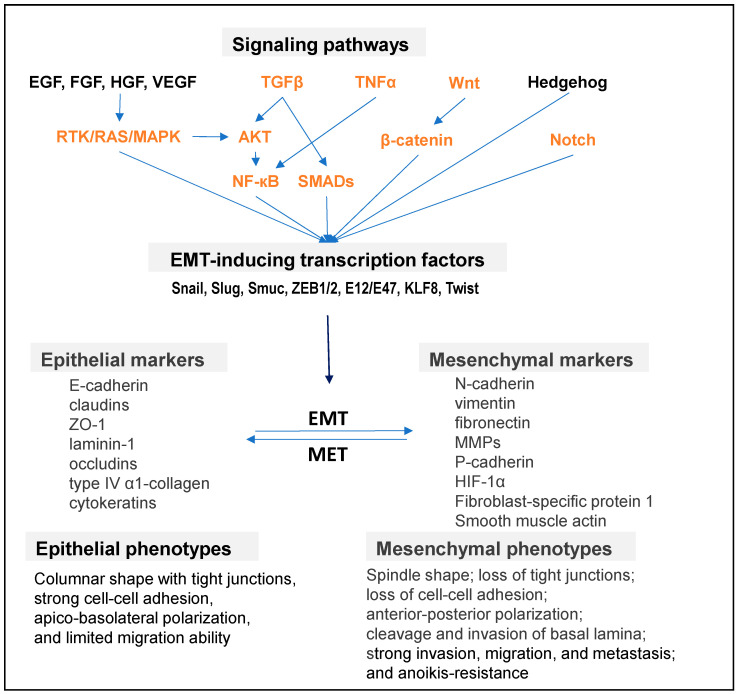
Major EMT-related signaling pathways that regulate EMT-TFs and markers of epithelial and mesenchymal cell states. The miR-181-regulated pathways are highlighted in orange. EMT, epithelial–mesenchymal transition; MET, mesenchymal-to-epithelial transition.

## Data Availability

No new data were created or analysed in this study. Data sharing is not applicable to this article.

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
