# Peer review of "The Multifaceted Roles of MicroRNA-181 in Stem Cell Differentiation and Cancer Stem Cell Plasticity"

_cells, 2025, doi:10.3390/cells14020132_

Round 1
Reviewer 1 Report
Comments and Suggestions for Authors
The data provides an up-to-date overview of the regulatory functions of miR-181 in both human stem cell differentiation and the stemness of cancer cells. The review summarizes current studies from the literature and describe in uno manuscript the miR-181-mediated signaling pathways that drive stem cell differentiation and contribute to the development of cancer stem cells. Furthermore it describes how modulating miR-181 expression in the biological behaviors of cancer stem cells and the significantly improve cancer treatment outcomes. The challenges associated with miRNA-based therapies and the targeted delivery of treatments using nanotechnology-based systems have been presented.
The manuscript provides a new comprehensive overview of the role of miR-181, highlighting its involvement not only in cancer pathogenesis but also in the regulation of epithelial-mesenchymal transition. Other reviews have been presented singular aspect of the miR-181 role but they have never described its multiple functions in a singular manuscript.
Fig 1 could indicate the position of miR-181in EMT-related signaling pathways to visualize its role in the pathway.
Author Response
Comments: The data provides an up-to-date overview of the regulatory functions of miR-181 in both human stem cell differentiation and the stemness of cancer cells. The review summarizes current studies from the literature and describe in uno manuscript the miR-181-mediated signaling pathways that drive stem cell differentiation and contribute to the development of cancer stem cells. Furthermore, it describes how modulating miR-181 expression in the biological behaviors of cancer stem cells and the significantly improve cancer treatment outcomes. The challenges associated with miRNA-based therapies and the targeted delivery of treatments using nanotechnology-based systems have been presented.
The manuscript provides a new comprehensive overview of the role of miR-181, highlighting its involvement not only in cancer pathogenesis but also in the regulation of epithelial-mesenchymal transition. Other reviews have been presented singular aspect of the miR-181 role but they have never described its multiple functions in a singular manuscript.
Fig. 1 could indicate the position of miR-181in EMT-related signaling pathways to visualize its role in the pathway.
Response : Thank you for pointing this out. We have accordingly made the revision on Figure 1 accordantly, and marked the changes in red in lines 231-232 on page 5. We also made some changes to the journal abbreviation in the reference list.
Reviewer 2 Report
Comments and Suggestions for Authors
In the current manuscript, Yang et al. provides a comprehensive review of the multifaceted roles of microRNA-181 (miR-181) in stem cell differentiation and cancer stem cell (CSC) plasticity. The review article highlights the importance of miR-181 in regulating key signaling pathways involved in epithelial-mesenchymal transition (EMT) and stem cell fate, which are crucial for both normal stem cell function and cancer progression. The review summarizes current literature on how miR-181 modulates EMT-associated factors, influencing the stemness of CSCs and driving stem cell differentiation. The manuscript emphasizes the diverse functions of miR-181 across different cancers and its potential as a therapeutic target to improve cancer treatment outcomes. Finally, it discusses the therapeutic potential of targeting miR-181, including challenges in miRNA-based therapies and the use of nanotechnology for targeted delivery. I have several suggestions for the improvement:
- While reviewing the literature, it would be beneficial if the authors could clarify whether the experiments were conducted using gain-of-function (overexpression) or loss-of-function (LNA, miRNA sponge, or knockout) approaches. This clarification is crucial to determine whether miR-181 functions at the physiological or therapeutic level.
- I would recommend that the authors include an additional figure (as Figure 1) to illustrate the sequences of miR-181 family members. Interestingly, all four miR-181 family members have the same seed sequence. Thus, discussing why they regulate distinct target genes (line 54) will provide more depth to this article.
- Following point 2, the miR-181 family members (a/b/c/d) tested in those documents should all be annotated in Tables 1 and 3.
- Section 5 appears somewhat mixed. The challenges and solutions of miRNA-based therapies should be presented as separate sections with expanded content. Additionally, including information on current clinical trials and past failures would be helpful for readers interested in understanding the current progress of miRNA-based therapies compared to siRNA-based approaches.
Author Response
In the current manuscript, Yang et al. provides a comprehensive review of the multifaceted roles of microRNA-181 (miR-181) in stem cell differentiation and cancer stem cell (CSC) plasticity. The review article highlights the importance of miR-181 in regulating key signaling pathways involved in epithelial-mesenchymal transition (EMT) and stem cell fate, which are crucial for both normal stem cell function and cancer progression. The review summarizes current literature on how miR-181 modulates EMT-associated factors, influencing the stemness of CSCs and driving stem cell differentiation. The manuscript emphasizes the diverse functions of miR-181 across different cancers and its potential as a therapeutic target to improve cancer treatment outcomes. Finally, it discusses the therapeutic potential of targeting miR-181, including challenges in miRNA-based therapies and the use of nanotechnology for targeted delivery. I have several suggestions for the improvement:
- While reviewing the literature, it would be beneficial if the authors could clarify whether the experiments were conducted using gain-of-function (overexpression) or loss-of-function (LNA, miRNA sponge, or knockout) approaches. This clarification is crucial to determine whether miR-181 functions at the physiological or therapeutic level.
Response: Thank you for your suggestion. Most of the experiments were conducted using gain-of-function (overexpression), and a few studies using loss-of-function (LNA, miRNA sponge, or knockout) approaches for further validation. Both gain-of-function and loss-of-function studies are complementary and can be used to build a comprehensive understanding of miRNA functions at the physiological and therapeutic levels.
- I would recommend that the authors include an additional figure (as Figure 1) to illustrate the sequences of miR-181 family members. Interestingly, all four miR-181 family members have the same seed sequence. Thus, discussing why they regulate distinct target genes (line 54) will provide more depth to this article.
Response: Thank you for your helpful suggestion. We would like to inform you that a figure illustrating the same seed sequence shared by miR-181 family members has already been published by Indrieri, et al. (Int J Mol Sci. 2020 Mar 18;21(6):2092. doi: 10.3390/ijms21062092). The correction made in red in lines 52-54 on page 2.
3. Following point 2, the miR-181 family members (a/b/c/d) tested in those documents should all be annotated in Tables 1 and 3.
As suggested, the miR-181 family members (a/b/c/d) tested are annotated respectively in Tables 1 and 2.
4. Section 5 appears somewhat mixed. The challenges and solutions of miRNA-based therapies should be presented as separate sections with expanded content. Additionally, including information on current clinical trials and past failures would be helpful for readers interested in understanding the current progress of miRNA-based therapies compared to siRNA-based approaches.
Response: Thank you for your helpful suggestion. As suggested, we separated the challenges and solutions of miRNA-based therapies from section 5 and put as section 6 with expanded content including the information on current clinical trials (see lines 491-505) on page 11.